# CONSENSUS ENERGY MINIMIZATION: ENSURING RELIABLE CONVERGENCE IN COLLABORATIVE DELIBERATION

## ABSTRACT

Multi-round deliberation among heterogeneous agents—whether humans, AI systems, or domain experts—offers opportunities to reduce diagnostic uncertainty through complementary reasoning. Yet such collaboration can also amplify errors if agents prematurely converge on unreliable conclusions. We propose a lightweight monitoring framework, Consensus Energy Minimization (CEM), that regulates collaborative decision-making without requiring domain-specific supervision. CEM formalizes deliberation as a dynamical system, where a confusion-aware consensus energy functional tracks both disagreement and convergence in low-reliability regions. The monitor applies stopping-time rules to either halt, continue, or steer discussion toward an agent's local expertise, ensuring convergence to high-confidence consensus. We provide theoretical guarantees showing that, under mild reliability assumptions, CEM provably avoids harmful convergence and achieves stability in safe consensus regions. Empirically, we demonstrate the framework on synthetic and real-world classification tasks, where CEM reduces uncertainty and improves joint accuracy across diverse interaction scenarios (ideal, asymmetric, and noisy). Our results highlight that principled monitoring, rather than model accuracy alone, is key to harnessing the benefits of deliberation.

## 1 INTRODUCTION

As AI becomes increasingly integrated into professional domains, research has shifted from technology-centered development toward collaborative designs that leverage diverse agents. Deliberation, characterized by thoughtful and reasoned discussion, plays a pivotal role in enabling constructive discourse and consensus-building across contexts (Bächtiger & Parkinson, 2019). In particular, deliberation among heterogeneous agents, including humans, AI systems, and domain experts, is increasingly adopted in high-stakes domains such as medicine, law, and scientific discovery (Zöller et al., 2024; Ma et al., 2025; Wang et al., 2025; Green & Chen, 2019).

While multi-agent deliberation can reduce uncertainty by combining complementary reasoning processes, it also introduces risks. Premature convergence on unreliable conclusions can amplify errors rather than mitigate them. Social psychology research has shown that repeated discussions may lead to group polarization, causing decisions to shift toward more extreme outcomes (Bang & Frith, 2017). Similar issues have emerged in AI multi-agent frameworks. Multiple LLM agents can prematurely converge to a consensus without sufficient critical evaluation, a phenomenon known as silent consensus (Wang et al., 2025). For example, Wang et al. (Wang et al., 2025) introduced "catfish agents" to inject structured dissent and disrupt premature consensus, Vodrahalli et al. (Vodrahalli et al., 2022) showed that even uncalibrated models can shape human reliance on AI advice, Carroll et al. (Carroll et al., 2020) explicitly modeled human behavior to improve coordination, Corvelo Benz and Gomez-Rodriguez (Benz & Rodriguez, 2024) proposed human-aligned calibration to better match AI confidence with human decision-making processes, and Cui et al. (Cui et al., 2025) designed consensus-free debate protocols (Free-MAD) that evaluate reasoning trajectories rather than relying solely on final majority votes. Despite these advances, most research on human–AI collaboration and multi-expert aggregation has focused on single-shot interactions or static accuracy improvements, leaving the iterative dynamics of multi-round deliberation relatively underexplored.

In this study, we propose a new framework for regulating collaborative deliberation through *Consensus Energy Minimization* (CEM). Our central idea is to treat agent interaction as an iterative dynamical system in the joint space of predictions and justifications. Supervisory mechanisms have been shown to improve both reasoning reliability and feedback quality. For instance, confidence-based filtering can halt low-quality reasoning traces to enhance accuracy (Fu et al., 2025), while annotation monitoring and incentive designs ensure consistency in human feedback (Liu et al., 2025). Inspired by these findings, we introduce a lightweight monitoring mechanism—a *deliberation monitor*—that does not solve the decision task directly but instead tracks the trajectory of collaboration. The monitor evaluates a confusion-aware *consensus energy functional* capturing two key risks: (i) persistent disagreement, indicating unresolved conflict, and (ii) low-confidence convergence, indicating fragile agreement in unreliable regions. Based on this energy, the monitor applies stopping-time rules to regulate deliberation by choosing one of three actions: STOP, halting collaboration when unsafe convergence is detected; STEER, guiding the discussion toward a more reliable agent's expertise; or CONTINUE, allowing further deliberation when convergence is safely emerging.

This formulation yields two benefits. First, it provides a principled account of multi-round deliberation: we show that under mild assumptions, consensus energy decreases monotonically and deliberation converges to high-confidence regions, while harmful consensus is detectable and avoidable. Second, it enables practical algorithms that require only historical confusion matrices and observed justifications, without domain-specific supervision or large-scale retraining.

In this paper, we focus on the two-agent case for clarity, analyzing the interaction between a pair of heterogeneous agents and demonstrating both theoretical guarantees and empirical behavior. Nonetheless, the framework naturally extends to multi-agent settings by aggregating divergence and reliability measures across agents, making CEM a general approach to safe collaborative reasoning. Through synthetic and real-world classification tasks, we show that CEM improves joint accuracy while reducing uncertainty across ideal, asymmetric, and noisy scenarios.

## 2 METHOD

We propose the *Consensus Energy Minimization (CEM)* framework (Figure 1) for regulating multi-round deliberation between agents. The framework formalizes interaction as an iterative dynamical system in which a lightweight monitor tracks predictions, justifications, and reliabilities, without directly solving the underlying task. The monitor computes a confusion-aware *consensus energy functional* that penalizes both disagreement and convergence in low-reliability regions, and applies stopping-time rules to decide when to halt, steer, or continue deliberation.

### 2.1 SETTING AND AGENTS

We consider a classification task with $K$ possible labels $y \in \{1, \ldots, K\}$ and define three primary roles in the system:

- **Agent H (human or human-like)**: This agent generates predictions $\hat{y}_H^{(t)}$ and optional feature-importance weights $w_H^{(t)}$ at each round $t$.

- **Agent L (AI or model)**: This agent produces predictions $\hat{y}_L^{(t)}$ and corresponding feature-importance weights $w_L^{(t)}$.

- **Monitor $M$**: Observes the sequence $\{(\hat{y}_H^{(t)}, w_H^{(t)}), (\hat{y}_L^{(t)}, w_L^{(t)})\}_{t=1}^T$ and controls the deliberation. The monitor does not make predictions itself; its role is to track the consensus dynamics and apply stopping or steering decisions.

### 2.2 CONFUSION-MATRIX–BASED RELIABILITY

To quantify each agent's expertise, we characterize them by their confusion matrices. For an agent $a \in \{H, L\}$, its confusion matrix $C_a$ is defined as:

$$C_a[i,j] = P(\hat{y}_a = j \mid y = i),$$

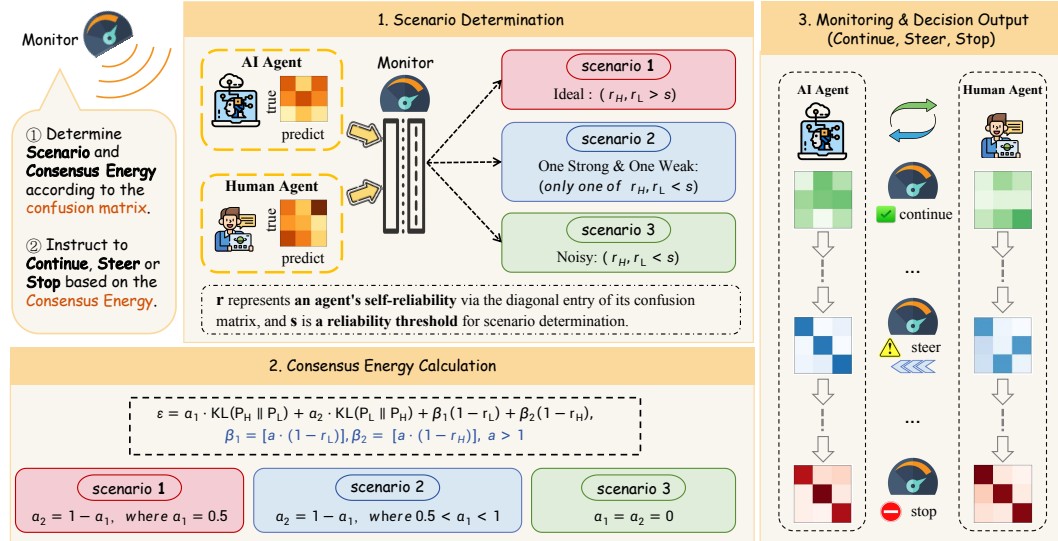

Figure 1: The Consensus Energy Minimization (CEM) framework for collaborative deliberation. The monitor first determines the interaction scenario based on agent reliabilities ($r_H, r_L$ relative to threshold $s$), then computes a confusion-aware consensus energy functional $\varepsilon$ that combines KL divergence terms with reliability penalties, and finally outputs one of three control decisions: Continue, Steer, or Stop, to ensure safe convergence in multi-round deliberation.

which encodes the probability that the agent predicts class $j$ when the true label is $i$. In particular, the diagonal entries $C_a[i, i]$ capture the agent's reliability on class $i$, and the off-diagonals capture its systematic biases. In practice, $C_a$ can be estimated from historical data or a calibration set.

From the confusion matrix, we derive two key quantities for each agent at each round $t$:

- **Reliability-informed posterior**: Given the agent's current prediction $\hat{y}_a^{(t)}$, we compute the posterior distribution over the true label:

$$P_a(y \mid \hat{y}_a^{(t)}) \propto C_a[y, \hat{y}_a^{(t)}].$$

  This reflects how trustworthy the prediction is, based on the agent's historical performance.

- **Self-reliability score**: We define the agent's instantaneous reliability as the diagonal entry corresponding to its current prediction:

$$r_a^{(t)} = C_a[\hat{y}_a^{(t)}, \hat{y}_a^{(t)}].$$

  The scalar $r_a^{(t)}$ is the probability that agent $a$'s prediction is correct, given historical performance. A high $r_a^{(t)}$ means agent $a$ is usually correct when it predicts this class, whereas a low $r_a^{(t)}$ signals caution. In summary, the pair $(P_a(\cdot \mid \hat{y}_a^{(t)}), r_a^{(t)})$ captures agent $a$'s belief and confidence at round $t$.

## 2.3 CONSENSUS ENERGY FUNCTIONAL

We now construct a *consensus energy* $\varepsilon^{(t)}$ to measure the quality of agreement between the agents at round $t$. Let $P_H^{(t)}(y) = P_H(y \mid \hat{y}_H^{(t)})$ and $P_L^{(t)}(y) = P_L(y \mid \hat{y}_L^{(t)})$ be the two agents' posteriors over the true class (as defined above). We define

$$\varepsilon^{(t)} = \alpha_1 D_{KL}\big(P_H^{(t)} \parallel P_L^{(t)}\big) + \alpha_2 D_{KL}\big(P_L^{(t)} \parallel P_H^{(t)}\big) \quad + \beta_1\big(1 - r_L^{(t)}\big) + \beta_2\big(1 - r_H^{(t)}\big).$$

Here $\alpha_1, \alpha_2, \beta_1, \beta_2 \geq 0$ are coefficients (which in principle can be adapted based on the agents' current reliabilities). Intuitively, the first two KL-divergence terms penalize persistent disagreement between the agents: they grow large if $P_H^{(t)}$ and $P_L^{(t)}$ differ significantly. The latter terms penalize

consensus in low-confidence regimes: if either agent has a low self-reliability $r_a^{(t)}$, then $(1 - r_a^{(t)})$ is large, raising $\varepsilon^{(t)}$. Thus, even when the agents' point predictions agree, the energy remains high if that agreement occurs in a region where an agent is known to be unreliable. One can set the weights so that, for example, a lower $r_L^{(t)}$ increases the penalty on agreement more strongly. A small $\varepsilon^{(t)}$ indicates that both agents are aligning on an answer they trust, whereas a large $\varepsilon^{(t)}$ signals either a lack of consensus or a potentially dangerous consensus by uncertain agents.

## 2.4 Monitoring and Stopping-Time Control

CEM treats deliberation as an iterative process of energy minimization. At each round $t$, the monitor performs the following three steps:

- **Scenario Determination.** The monitor compares each agent's self-reliability $r_H^{(t)}, r_L^{(t)}$ against a preset threshold $s$ to categorize the interaction. For example, if both $r_H^{(t)}$ and $r_L^{(t)}$ exceed $s$, we have an *ideal* scenario (both agents are knowledgeable); if exactly one exceeds $s$, we have an *asymmetric* scenario (one strong agent, one weak agent); and if both fall below $s$, the scenario is *noisy* (neither is reliable). This classification helps interpret the energy dynamics.

- **Consensus Energy Calculation.** Compute the current energy $\varepsilon^{(t)}$ as defined above, using the agents' posteriors and reliabilities. This tracks the evolution of disagreement and confidence in the discussion.

- **Decision Output.** Based on $\varepsilon^{(t)}$ and its recent change $\Delta \varepsilon^{(t)} = \varepsilon^{(t)} - \varepsilon^{(t-1)}$, the monitor issues one of three instructions:

  - **CONTINUE** if $\varepsilon^{(t)}$ is steadily decreasing and still above a small safety threshold $\epsilon$. This means the agents are safely moving toward consensus, so the discussion can proceed.

  - **STEER** if deliberation has stalled in an *asymmetric* scenario (one agent is much more reliable than the other). In this case, the monitor will guide the weaker agent's reasoning toward the stronger agent's expertise (see below).

  - **STOP** if $\varepsilon^{(t)}$ remains high (above a low-confidence cutoff $\epsilon_{\text{low}}$) without decreasing. This indicates that continued discussion is reinforcing an unreliable consensus, so we halt to avoid harmful convergence.

Formally, we can define the stopping time $\tau^*$ as the earliest round $t$ such that either $\varepsilon^{(t)} \leq \epsilon$ (safe consensus achieved) or the energy descent has stagnated above the low threshold:

$$\tau^* = \min\Big\{ t \mid \varepsilon^{(t)} \leq \epsilon \text{ or } \big[\Delta\varepsilon^{(t)} > -\delta \text{ and } \varepsilon^{(t)} > \epsilon_{\text{low}}\big] \Big\}. \tag{1}$$

Here $\epsilon > 0$ is the convergence threshold, $\epsilon_{\text{low}} > 0$ is the confidence cutoff, and $\delta > 0$ detects stagnation in the energy decrease. By this rule, deliberation stops as soon as the energy falls below $\epsilon$ (ensuring a high-confidence consensus) or if the energy has stopped decreasing while still above $\epsilon_{\text{low}}$ (preventing a low-confidence consensus). In our experiments, the default values are set as $\epsilon = 0.05$, $\epsilon_{\text{low}} = 0.3$, and $\delta = 0.01$.

## 2.5 Steering as Feature-Weight Optimization

When the monitor issues a **STEER** instruction, it adjusts the weaker agent's reasoning by modifying its feature-weight vector. Concretely, suppose agent $H$ is identified as weaker and agent $L$ is stronger at round $t$. Let $w_H^{(t)}$ be the weight vector of the weaker agent. We perform a small projected gradient step on a composite objective to update $w_H$:

$$w_H^{(t+1)} = w_H^{(t)} - \eta \nabla_{w_H} \big[\alpha \| w_H - w_H^{(t)} \|^2 + \beta \| w_H - w_{\text{expert}}^{(t)} \|^2 + \gamma E(w_H, w_L^{(t)})\big], \tag{2}$$

Here $\eta > 0$ is a small step size, and $w_{\text{expert}} = w_L^{(t)}$ is the weight vector of the more reliable agent (serving as a reference). The objective inside the gradient has three terms: (1) $\| w_H - w_H^{(t)} \|^2$ penalizes large changes (preserving interpretability and consistency), (2) $\| w_H - w_{\text{expert}} \|^2$ encourages $H$'s weights to move closer to the stronger agent's weights (aligning their explanations), and (3)

$\varepsilon(w_H, w_L^{(t)})$ is the consensus energy as a function of $w_H$ (promoting explicit reduction of disagreement). In effect, this update nudges the weaker agent's justification toward the expert's perspective while reducing the energy. Importantly, this steering acts on the explanation (feature weights) and does not change the agent's underlying predictive model.

## 2.6 EXTENSIONS

CEM admits several natural extensions:

- **Dynamic reliability:** If the human agent $H$ is learning over time, one could update $C_H$ online as more data or feedback becomes available.
- **Adaptive thresholds:** The thresholds $\epsilon, \delta$ could be tuned dynamically (for instance via reinforcement learning) based on observed outcomes or task requirements.
- **Energy visualization:** One might visualize the energy landscape of a task by plotting $\varepsilon$ over deliberation trajectories, helping to interpret where and why discussion stalls.

**Multi-Agent Extension.** Although we analyze the two-agent case for clarity, the framework extends naturally to $N$ agents. Define

$$\varepsilon^{(t)} = \boldsymbol{\alpha} \sum_{i \neq j} D_{KL}(P_i^{(t)} \parallel P_j^{(t)}) + \boldsymbol{\beta} \sum_i (1 - r_i^{(t)}).$$

The same stopping-time and steering logic applies: consensus is considered safe when pairwise disagreements vanish and at least one agent maintains bounded reliability. Monotonicity arguments extend by convexity, ensuring that reliable subgroups dominate the consensus trajectory, while groups with uniformly low reliability are halted. This makes CEM a general approach to multi-agent deliberation, covering teams, panels, or ensembles of models.

## 3 THEORETICAL GUARANTEES

We now formalize the guarantees of the Consensus Energy Minimization framework. The following results establish that CEM ensures safe convergence and prevents harmful consensus. We work under the standing assumptions summarized in Appendix B (finite label set, smoothed/confidence-calibrated $C_a$ with positive support, closed convex $\mathcal{W}$, $\varepsilon(\cdot, w_L)$ being $C^1$ and $L$-smooth in $w_H$, and projected steps with $\eta < 2/L$); complete proofs are given in Appendix B.2.

**Lemma 1** (Non-negativity and Monotonicity). *Let the consensus energy at round $t$ be defined as in Secti2. Under the technical conditions specified in Appendix B, and with gradient-based* STEER *updates using step size $0 < \eta < \eta_{\max}$, the energy sequence satisfies:*

$$\varepsilon^{(t+1)} \leq \varepsilon^{(t)}, \quad \forall t,$$

*and remains nonnegative.*

*Proof sketch.* Nonnegativity follows from the non-negativity of KL divergence and reliability terms. The monotonicity is guaranteed by the descent properties of the projected gradient method applied to the smooth consensus energy functional. See Appendix B.2 for the complete proof. $\square$

**Theorem 1** (Safe Convergence under Bounded Reliability). *Suppose at least one agent maintains reliability bounded away from zero, i.e. $\exists a \in \{H, L\}$ with $r_a^{(t)} \geq \rho > 0$ for all $t$. Then under the assumptions in Appendix B, the deliberation process either:*

- *Halts at finite time $\tau^*$ with $\varepsilon^{(\tau^*)} \leq \epsilon$, or*

- *Converges to a consensus state where $\lim_{t \to \infty} \varepsilon^{(t)} \leq \epsilon$.*

*Proof sketch.* By Lemma 1, $\{\varepsilon^{(t)}\}$ is nonincreasing and bounded below, hence convergent. The bounded reliability ensures that the system cannot persist at stationary limits with nonvanishing pairwise disagreement (KL contributions would keep decreasing the energy). The complete convergence analysis is provided in Appendix B.2. In particular, if some agent $a$ maintains $r_a^{(t)} \geq \rho > 0$, choosing $\epsilon > \beta_{\neg a}(1 - \rho)$ guarantees termination in the safe region (Appendix B.2). $\square$

**Theorem 2** (Detectability of Harmful Convergence). *If both agents' reliabilities vanish ($r_H^{(t)}, r_L^{(t)} \to 0$), then under the conditions in Appendix B and for any $\epsilon_{\text{low}} < \beta_1 + \beta_2$:*

$$\Delta\varepsilon^{(t)} \to 0, \quad \varepsilon^{(t)} > \epsilon_{\text{low}} \text{ for sufficiently large } t.$$

*Consequently, the* STOP *condition in Eq. 1 is triggered, preventing reinforcement of unreliable consensus.*

*Proof sketch.* Vanishing reliabilities cause the penalty term to approach $\beta_1 + \beta_2$, keeping the energy bounded away from zero unless disagreement disappears; thus for any $\epsilon_{\text{low}} < \beta_1 + \beta_2$ the inequality holds eventually. Meanwhile, gradient magnitudes diminish and the one-step decrease vanishes, i.e., $\Delta\varepsilon^{(t)} \to 0$. The stopping condition in Eq. 1 detects stagnation above $\epsilon_{\text{low}}$ and halts. See Appendix B.2. □

**Corollary 1** (Multi-Agent Extension). *For $N$ agents with the extended energy functional, the monotonicity, convergence, and detectability guarantees extend naturally under similar technical conditions.*

## 4 EXPERIMENTS

### 4.1 EXPERIMENTAL DESIGN

We evaluate our framework on one **synthetic dataset** and three real-world benchmarks (**Drug Classification**, **Weather Type Classification**, and **Customer Segmentation**) with dataset details in Appendix C. To simulate both human and agent behaviors, Large language models (LLMs) are employed, with prompts and configurations provided in Appendix D. As detailed in the Method section, we categorize collaboration into three scenarios: *Ideal*, *One Strong & One Weak*, and *Noisy*, based on the reliability scores $r_H$ and $r_L$. We then systematically examine how these scenarios influence model performance, focusing on two key metrics: (i) absolute accuracy of final decisions for each agent and (ii) the consensus energy value, where lower energy indicates safer convergence with reduced uncertainty.

#### 4.1.1 SCENARIO DESIGN

**Scenario1: Ideal** In the ideal setting, both agents maintain consistently high reliability ($r_H, r_L > s$). This scenario evaluates whether the framework can sustain deliberation toward a safe consensus without unnecessary intervention. Since such an ideal case is rare in real-world applications, we conducted experiments on a synthetic dataset.

**Scenario2: One Strong & One Weak** In the One Strong & One Weak scenario, reliability is asymmetric ($r_H, r_L > s$ for only one agent), meaning the stronger agent compensates for the weaker agent's shortcomings. We evaluate whether the framework can steer deliberation to align the weaker agent with the stronger one without misleading the latter. Experiments were conducted on both a synthetic dataset and a real-world drug classification dataset.

**Scenario 3: Noisy** In the Noisy scenario, both agents exhibit low reliability ($r_H, r_L \to 0$), with additional perturbations injected into their justifications. We investigate whether the stopping rule can prevent harmful error reinforcement in the **non-complementary** case, and whether the consensus energy functional can still drive safe convergence when agents are **complementary** despite lacking bounded reliability. Experiments were conducted on synthetic data and two real-world datasets: weather type classification (complementary) and customer segmentation (non-complementary).

#### 4.1.2 FUNDAMENTAL EXPERIMENTAL RESULTS

**Summary of Accuracy and Consensus Energy Trends** The accuracy and consensus energy trends of the agents across the three experimental scenarios on each dataset are shown in Figure 2.

- In Scenario 1, Both agents' accuracies steadily improve and approach 1, while consensus energy decreases, showing safe convergence and reduced uncertainty.

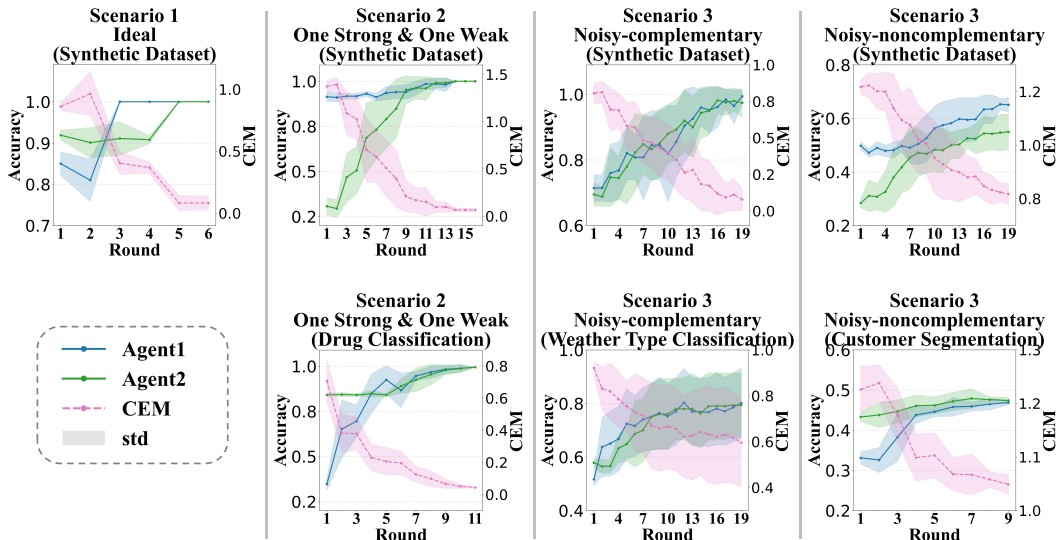

Figure 2: Accuracy and consensus energy trends of agents across three experimental scenarios.

- Scenario 2 features distinct initial accuracy conditions: one agent starting high (0.9) and the other low (0.3), the framework mitigates the negative impact of the less accurate agent, resulting in substantial accuracy gains for the latter and modest gains for the more accurate agent, with both eventually nearing accuracy 1 as consensus energy declines.

- Even under unbounded reliability setting in Scenario 3, both agents exhibit accuracy improvements. The complementary case outperforms the non-complementary by about 0.1, and the latter avoids convergence on errors, confirming the monitoring and termination mechanism's effectiveness.

**Stepwise Confusion Matrix Visualization**  Confusion matrix visualization provides insight into the distribution of agents' cognitive capabilities after autonomous learning or mutual interaction. It also reveals the influence of monitoring instructions on agent interaction. For each scenario and dataset, we randomly select one iteration of the confusion matrix for illustration. Figure 3 presents the two most contrasting cases on the synthetic dataset (the ideal and noisy-non-complementary scenarios), while the remaining five results are provided in Appendix E.

- In ideal case, both agents possess strong capabilities such that further self-learning yields limited improvement. In this setting, timely mutual interaction allows their strengths to complement each other, leading to enhanced overall performance.

- In noisy-non-complementary case, when both agents initially exhibit low capabilities, early-stage interference is minimized to enable autonomous learning and avoid premature consensus. As capabilities and reliability gradually improve, if one agent develops cognitive bias, timely "steer" instructions can guide it back to the correct trajectory, preventing harmful consensus even without further capability gains.

## 4.2 ABLATION STUDIES AND COMPARATIVE ANALYSIS

### 4.2.1 FREE DISCUSSION WITHOUT MONITOR

To evaluate the effectiveness of our framework in facilitating multi-agent learning and interaction, we conducted a comparative study between free discussion without monitor supervision (Free Discussion) and the complete framework (Figure 4). In the free discussion setting, agents, whose behaviors are simulated by Large Language Models (LLMs), exchange feedback and update their reasoning based on observed cases in each round, brainstorming collectively without a monitor to decide whether deliberation should continue, steer, or stop. This comparison illustrates the peak performance achievable through collaborative deliberation without monitoring. In contrast, the moni-

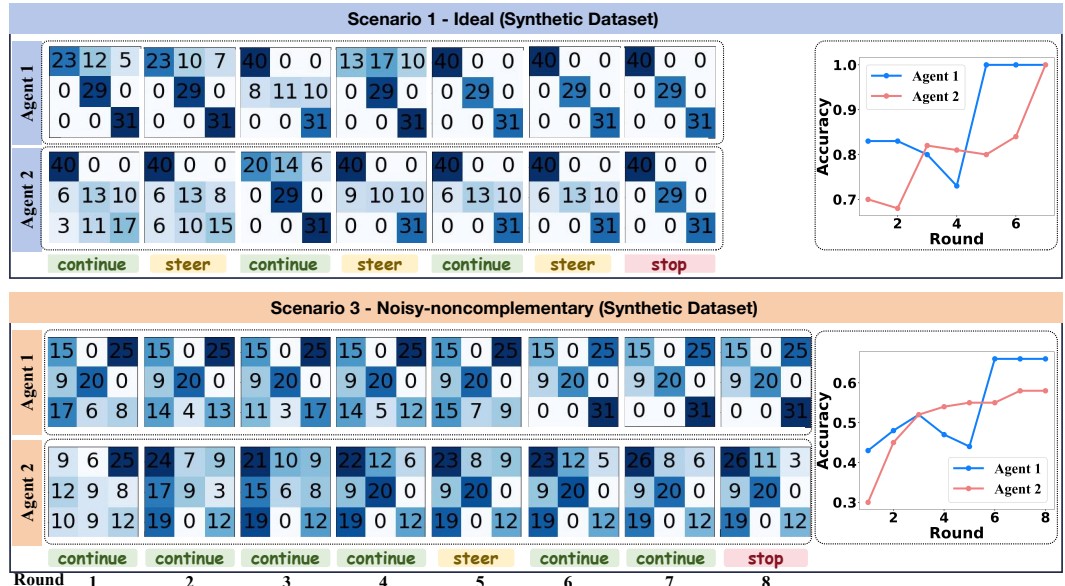

Figure 3: Stepwise confusion matrices and accuracy trends for two representative scenarios with monitor instructions.

tored framework achieves higher accuracy with fewer rounds, reducing unnecessary interaction and enhancing both the quality and stability of deliberation.

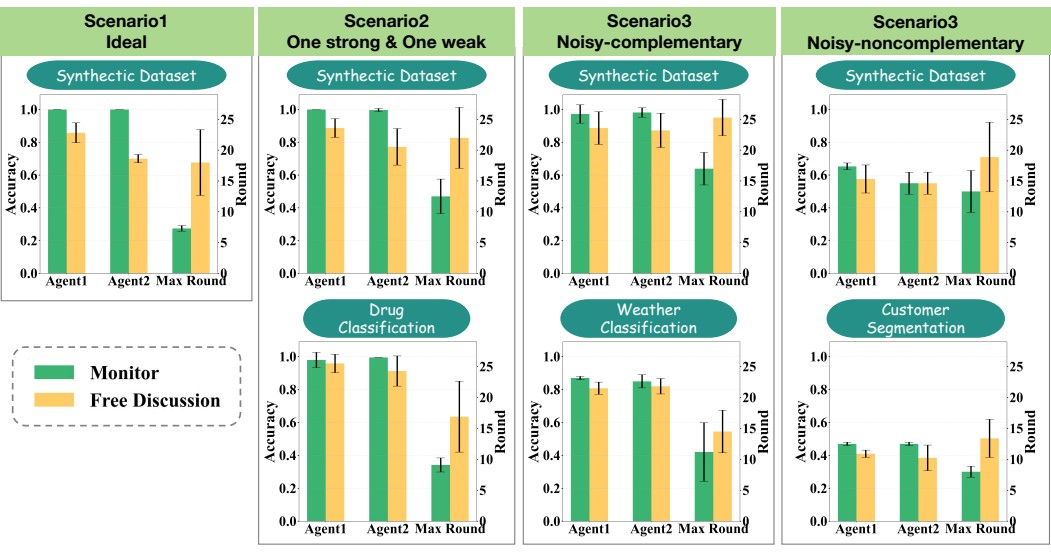

Figure 4: Performance comparison of accuracy and max round: monitor supervision vs. free discussion across three scenarios.

### 4.2.2 MODELING REALISTIC ACCEPTANCE HETEROGENEITY AMONG AGENTS

We evaluate the framework across multiple interaction rounds (4, 8, and 12) and acceptance levels (25%, 50%, 75%, 100%), reporting mean accuracies ± standard deviations for Agent 1 (human-simulating) and Agent 2 (LLM), along with consensus energy. Overall, accuracies consistently improve while consensus energy declines, reflecting effective steering and stopping mechanisms.

- In the "ideal" setting, Table 1 shows that when both agents resist deliberation (low acceptance rate), consensus energy gradually weakens. Even at a 25% acceptance rate, the framework remains effective: consensus energy decreases from $0.775 \pm 0.164$ to $0.478 \pm 0.167$ between rounds 4 and 12, with both agents improving in accuracy. One increases from $0.828 \pm 0.015$ to $0.860 \pm 0.080$, while the other shows a larger gain from $0.663 \pm 0.100$ to $0.836 \pm 0.074$.

- When reliability is asymmetric, accuracy gains shrink as the acceptance rate decreases but remain positive for both agents. Crucially, even when the stronger agent fully accepts weaker input, the framework prevents harmful convergence, ensuring the weaker agent improves more than the stronger one, demonstrating resilience to asymmetric influence and safeguarding against dangerous consensus.

- In noisy environments, complementary noise allows accuracy improvements at a 25% acceptance rate, though with smaller gains. In non-complementary noisy settings, the early-stopping mechanism halts deliberation before errors can reinforce, so ablation results are omitted.

Table 1: Ablation study under varying acceptance rates across three scenarios on both synthetic and real-world datasets.

**Scenario 1 – Ideal**

**Synthetic Dataset**

| Acceptance | Round=4 | | | Round=8 | | | Round=12 | | |
|---|---|---|---|---|---|---|---|---|---|
| | Agent1-Acc | Agent2-Acc | CEM | Agent1-Acc | Agent2-Acc | CEM | Agent1-Acc | Agent2-Acc | CEM |
| 25% | $0.828_{\pm0.015}$ | $0.663_{\pm0.100}$ | $0.775_{\pm0.164}$ | $0.834_{\pm0.020}$ | $0.762_{\pm0.119}$ | $0.627_{\pm0.162}$ | $0.860_{\pm0.080}$ | $0.836_{\pm0.074}$ | $0.478_{\pm0.167}$ |
| 50% | $0.831_{\pm0.020}$ | $0.747_{\pm0.126}$ | $0.634_{\pm0.212}$ | $0.907_{\pm0.069}$ | $0.823_{\pm0.083}$ | $0.445_{\pm0.205}$ | $0.878_{\pm0.068}$ | $0.868_{\pm0.080}$ | $0.412_{\pm0.139}$ |
| 75% | $0.835_{\pm0.030}$ | $0.691_{\pm0.100}$ | $0.689_{\pm0.145}$ | $0.925_{\pm0.075}$ | $0.870_{\pm0.090}$ | $0.319_{\pm0.100}$ | $1.000_{\pm0.000}$ | $1.000_{\pm0.000}$ | $0.034_{\pm0.018}$ |
| 100% | $0.825_{\pm0.040}$ | $0.814_{\pm0.030}$ | $0.529_{\pm0.050}$ | $1.000_{\pm0.000}$ | $1.000_{\pm0.000}$ | $0.038_{\pm0.015}$ | $1.000_{\pm0.000}$ | $1.000_{\pm0.000}$ | $0.038_{\pm0.015}$ |

**Scenario 2 – One Strong & One Weak**

**Synthetic Dataset**

| Acceptance | Round=4 | | | Round=8 | | | Round=12 | | |
|---|---|---|---|---|---|---|---|---|---|
| | Agent1-Acc | Agent2-Acc | CEM | Agent1-Acc | Agent2-Acc | CEM | Agent1-Acc | Agent2-Acc | CEM |
| 25% | $0.906_{\pm0.019}$ | $0.393_{\pm0.129}$ | $1.226_{\pm0.220}$ | $0.910_{\pm0.025}$ | $0.486_{\pm0.192}$ | $1.057_{\pm0.327}$ | $0.906_{\pm0.023}$ | $0.573_{\pm0.186}$ | $0.905_{\pm0.307}$ |
| 50% | $0.898_{\pm0.027}$ | $0.470_{\pm0.128}$ | $1.092_{\pm0.221}$ | $0.907_{\pm0.026}$ | $0.682_{\pm0.153}$ | $0.694_{\pm0.280}$ | $0.917_{\pm0.034}$ | $0.780_{\pm0.160}$ | $0.510_{\pm0.289}$ |
| 75% | $0.909_{\pm0.023}$ | $0.525_{\pm0.126}$ | $0.992_{\pm0.220}$ | $0.918_{\pm0.036}$ | $0.686_{\pm0.164}$ | $0.688_{\pm0.279}$ | $0.954_{\pm0.048}$ | $0.866_{\pm0.133}$ | $0.331_{\pm0.252}$ |
| 100% | $0.917_{\pm0.016}$ | $0.506_{\pm0.133}$ | $1.025_{\pm0.220}$ | $0.940_{\pm0.036}$ | $0.848_{\pm0.164}$ | $0.397_{\pm0.304}$ | $0.985_{\pm0.000}$ | $0.991_{\pm0.027}$ | $0.096_{\pm0.057}$ |

**Drug Classification**

| Acceptance | Round=4 | | | Round=8 | | | Round=12 | | |
|---|---|---|---|---|---|---|---|---|---|
| | Agent1-Acc | Agent2-Acc | CEM | Agent1-Acc | Agent2-Acc | CEM | Agent1-Acc | Agent2-Acc | CEM |
| 25% | $0.655_{\pm0.145}$ | $0.839_{\pm0.012}$ | $0.358_{\pm0.120}$ | $0.842_{\pm0.092}$ | $0.849_{\pm0.011}$ | $0.210_{\pm0.069}$ | $0.861_{\pm0.084}$ | $0.845_{\pm0.022}$ | $0.203_{\pm0.072}$ |
| 50% | $0.711_{\pm0.052}$ | $0.874_{\pm0.062}$ | $0.308_{\pm0.045}$ | $0.893_{\pm0.089}$ | $0.878_{\pm0.059}$ | $0.162_{\pm0.063}$ | $0.911_{\pm0.097}$ | $0.913_{\pm0.062}$ | $0.133_{\pm0.074}$ |
| 75% | $0.698_{\pm0.092}$ | $0.852_{\pm0.049}$ | $0.321_{\pm0.086}$ | $0.907_{\pm0.086}$ | $0.880_{\pm0.055}$ | $0.153_{\pm0.059}$ | $0.947_{\pm0.054}$ | $0.936_{\pm0.066}$ | $0.101_{\pm0.056}$ |
| 100% | $0.849_{\pm0.109}$ | $0.848_{\pm0.017}$ | $0.230_{\pm0.083}$ | $0.968_{\pm0.034}$ | $0.954_{\pm0.057}$ | $0.099_{\pm0.049}$ | $0.995_{\pm0.000}$ | $0.995_{\pm0.000}$ | $0.044_{\pm0.005}$ |

**Scenario 3 – Noisy-complementary**

**Synthetic Dataset**

| Acceptance | Round=4 | | | Round=8 | | | Round=12 | | |
|---|---|---|---|---|---|---|---|---|---|
| | Agent1-Acc | Agent2-Acc | CEM | Agent1-Acc | Agent2-Acc | CEM | Agent1-Acc | Agent2-Acc | CEM |
| 25% | $0.721_{\pm0.052}$ | $0.700_{\pm0.081}$ | $0.741_{\pm0.100}$ | $0.756_{\pm0.073}$ | $0.750_{\pm0.109}$ | $0.652_{\pm0.129}$ | $0.812_{\pm0.114}$ | $0.755_{\pm0.082}$ | $0.594_{\pm0.169}$ |
| 50% | $0.729_{\pm0.070}$ | $0.692_{\pm0.104}$ | $0.747_{\pm0.136}$ | $0.814_{\pm0.114}$ | $0.764_{\pm0.125}$ | $0.533_{\pm0.205}$ | $0.812_{\pm0.130}$ | $0.834_{\pm0.168}$ | $0.414_{\pm0.244}$ |
| 75% | $0.731_{\pm0.057}$ | $0.736_{\pm0.100}$ | $0.731_{\pm0.113}$ | $0.770_{\pm0.117}$ | $0.807_{\pm0.118}$ | $0.575_{\pm0.164}$ | $0.844_{\pm0.132}$ | $0.848_{\pm0.109}$ | $0.375_{\pm0.185}$ |
| 100% | $0.768_{\pm0.088}$ | $0.745_{\pm0.065}$ | $0.689_{\pm0.120}$ | $0.845_{\pm0.105}$ | $0.833_{\pm0.097}$ | $0.469_{\pm0.183}$ | $0.905_{\pm0.110}$ | $0.920_{\pm0.094}$ | $0.262_{\pm0.175}$ |

**Weather Type Classification**

| Acceptance | Round=4 | | | Round=8 | | | Round=12 | | |
|---|---|---|---|---|---|---|---|---|---|
| | Agent1-Acc | Agent2-Acc | CEM | Agent1-Acc | Agent2-Acc | CEM | Agent1-Acc | Agent2-Acc | CEM |
| 25% | $0.590_{\pm0.089}$ | $0.574_{\pm0.014}$ | $0.876_{\pm0.094}$ | $0.669_{\pm0.131}$ | $0.587_{\pm0.053}$ | $0.806_{\pm0.131}$ | $0.688_{\pm0.120}$ | $0.621_{\pm0.104}$ | $0.772_{\pm0.155}$ |
| 50% | $0.638_{\pm0.144}$ | $0.588_{\pm0.052}$ | $0.833_{\pm0.140}$ | $0.740_{\pm0.149}$ | $0.643_{\pm0.106}$ | $0.726_{\pm0.163}$ | $0.768_{\pm0.135}$ | $0.735_{\pm0.141}$ | $0.642_{\pm0.207}$ |
| 75% | $0.692_{\pm0.109}$ | $0.669_{\pm0.116}$ | $0.727_{\pm0.199}$ | $0.740_{\pm0.138}$ | $0.736_{\pm0.121}$ | $0.667_{\pm0.210}$ | $0.772_{\pm0.140}$ | $0.776_{\pm0.135}$ | $0.595_{\pm0.232}$ |
| 100% | $0.668_{\pm0.121}$ | $0.633_{\pm0.053}$ | $0.785_{\pm0.123}$ | $0.751_{\pm0.125}$ | $0.750_{\pm0.124}$ | $0.672_{\pm0.187}$ | $0.803_{\pm0.128}$ | $0.780_{\pm0.116}$ | $0.622_{\pm0.179}$ |

## 5 CONCLUSION

In this work, we introduced Consensus Energy Minimization (CEM), a lightweight monitoring framework for regulating multi-round collaborative deliberation among heterogeneous agents. By modeling deliberation as a dynamical system, CEM employs a confusion-aware consensus energy functional that penalizes both persistent disagreement and convergence in low-reliability regions. Through theoretical analysis and empirical evaluation, we demonstrated that principled monitoring, rather than accuracy alone, is essential for preventing harmful consensus, establishing CEM as a general foundation for reliable human–AI collaboration and a promising direction for future applications in high-stakes decision-making.

REPRODUCIBILITY STATEMENT

All codes and datasets used in this work will be made publicly available upon acceptance.

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

# A  RELATED WORK

**Cognitive Alignment**  A rich vein of research has shown that effective Human-AI collaboration system requires both human and AI agents to progressively converge toward shared beliefs and judgments (Noti et al., 2025; Tian et al., 2023; Liu et al., 2025; Wang et al., 2020). The need for AI to integrate into human workflows and co-manage tasks with mutual understanding is central to creating effective Human-AI collaboration(Wang et al., 2020). For example, Theory of Mind models show how agents can infer and adapt to others' mental states (Rabinowitz et al., 2018), while studies on human learning dynamics highlight that both parties update their internal models through interaction (Tian et al., 2023). In AI-assisted decision-making, algorithms must not only provide immediately useful advice but also foster long-term improvements in human reasoning across rounds (Noti et al., 2025). Moreover, the reliability of human feedback plays a critical role in shaping the convergence outcome (Liu et al., 2025). In clinical domains, cognitive informatics stresses the need for alignment with physicians' reasoning processes (Patel & Kannampallil, 2015). Taken together, these studies suggest that human–AI collaboration should be understood as a multi-round convergence process, where mutual adaptation gradually reduces disagreement and improves joint accuracy.

**Expert modeling**  Expert modeling has been extensively studied in medical diagnosis and decision-making, focusing on annotator reliability, inter-observer variability, and human–AI complementarity (Raykar et al., 2010; Steyvers et al., 2022; Noti et al., 2025; Gebeşçe et al., 2025). Prior work has mainly relied on Bayesian aggregation of noisy labels (Raykar et al., 2010; Steyvers et al., 2022) or divergence-based measures of decision similarity and trust (Gebeşçe et al., 2025). These approaches advance statistical modeling but typically abstract expertise into latent reliability parameters, limiting interpretability and failing to capture domain-specific specialization. Recent work on AI-assisted decision-making has emphasized the importance of adapting the algorithm's feature selection process based on the evolving expertise and understanding of human decision-makers(Steyvers et al., 2022). In this study, we model two heterogeneous agents with varying expertise levels, where their interactions lead to an evolving understanding of the decision-making rules, guided by the *Consensus Energy Minimization* (CEM) framework.

**Consensus Energy Minimization**  The idea of consensus energy builds on energy-based views of uncertainty reduction. The Free Energy Principle suggests agents minimize free energy to align beliefs with reality (Mazzaglia et al., 2022). Perdomo et al. showed that model predictions can shift the data distribution and demonstrated that learning algorithms can converge to a performatively stable point by iteratively retraining (Perdomo et al., 2021). Moreover, Agarwal and Brown found that by combining historical behavior with current states, model can effectively avoid feedback loops and stabilize long-term performance (Agarwal & Brown, 2024). Recent methods such as DeepConf prune low-confidence reasoning to enforce internal consistency (Fu et al., 2025). Extending these ideas, our CEM framework formalizes a consensus energy between human and AI latent states. By minimizing it online with confusion-matrix–based monitoring, CEM enables cooperative alignment that improves accuracy and stability in multi-round interaction.

# B  TECHNICAL ASSUMPTIONS AND DETAILED PROOFS

## B.1  TECHNICAL ASSUMPTIONS

The theoretical guarantees rely on the following technical assumptions:

**Assumption 1** (Decision Consistency)**.**  *1. **Rational Alignment:** Each agent's feature weights $w_a^{(t)}$ accurately reflect their internal belief state and are consistent with their reliability-informed posterior $P_a^{(t)}$*

    *2. **Truthful Reporting:** Agents report their genuine assessments without strategic manipulation or systematic bias between internal reasoning and external expression*

    *3. **Cognitive Transparency:** The mapping from internal feature importance to external predictions is well-defined and stable over time*

**Assumption 2** (Smoothness and Regularity). *1. The confusion matrices $C_a$ are strictly positive definite:* $\min_i C_a[i,i] \geq \gamma > 0$ *for some* $\gamma > 0$

2. *The feature weight vectors* $w_a^{(t)}$ *belong to a compact convex set* $\mathcal{W} \subset \mathbb{R}^d$

3. *The consensus energy functional* $\varepsilon(w_H, w_L)$ *is continuously differentiable and $L$-smooth in* $w_H$ *on* $\mathcal{W}$

4. *The step size satisfies* $0 < \eta < 2/L$ *for projected gradient descent*

**Assumption 3** (Bounded Interactions). *1. The label space is finite:* $|\mathcal{Y}| = K < \infty$

2. *The reliability scores are bounded:* $r_a^{(t)} \in [0,1]$ *for all* $a, t$

3. *The energy coefficients are positive:* $\alpha_1, \alpha_2, \beta_1, \beta_2 > 0$

## B.2  DETAILED PROOFS

*Proof of Lemma 1.* We prove the two claims separately:

**Nonnegativity:** The consensus energy $\varepsilon^{(t)}$ consists of four non-negative terms:

- KL divergences: $D_{KL}(P \parallel Q) \geq 0$ for any distributions

- Reliability penalties: $1 - r_a^{(t)} \geq 0$ since $r_a^{(t)} \in [0,1]$

Since all coefficients $\alpha_1, \alpha_2, \beta_1, \beta_2$ are positive, $\varepsilon^{(t)}$ is a sum of non-negative terms, hence $\varepsilon^{(t)} \geq 0$.

**Monotonicity:** We consider two cases:

*Case 1: Steering is applied.* When the monitor issues a STEER instruction, it updates the weaker agent's feature weights $w_H$ to minimize a composite objective:

$$\mathcal{L}(w_H) = \underbrace{\alpha \|w_H - w_H^{(t)}\|^2}_{\text{stability}} + \underbrace{\beta \|w_H - w_{\text{expert}}^{(t)}\|^2}_{\text{alignment}} + \underbrace{\gamma \varepsilon(w_H, w_L^{(t)})}_{\text{consensus}}$$

By Assumption 2.3, $\mathcal{L}$ is smooth, so gradient descent with step size $\eta < 2/L$ guarantees:

$$\mathcal{L}(w_H^{(t+1)}) \leq \mathcal{L}(w_H^{(t)}) - \frac{\eta}{2} \|\nabla \mathcal{L}(w_H^{(t)})\|^2 \leq \mathcal{L}(w_H^{(t)})$$

Since $\varepsilon^{(t)}$ appears in $\mathcal{L}$ and the other terms are non-negative, we have:

$$\varepsilon^{(t+1)} \leq \mathcal{L}(w_H^{(t+1)}) \leq \mathcal{L}(w_H^{(t)}) \Rightarrow \varepsilon^{(t+1)} \leq \varepsilon^{(t)}$$

*Case 2: Natural deliberation.* When agents discuss without steering, we assume they rationally update their beliefs to reduce disagreement (e.g., through Bayesian updating or consensus-seeking behavior). This naturally decreases the KL divergence terms in $\varepsilon^{(t)}$, while reliability scores either improve or remain stable. Thus, $\varepsilon^{(t+1)} \leq \varepsilon^{(t)}$.

In both cases, the energy does not increase, proving monotonicity. $\qquad\square$

*Proof of Theorem 1.* By Lemma 1, $\{\varepsilon^{(t)}\}$ is nonincreasing and bounded below by 0, hence convergent. Let $\varepsilon^* = \lim_{t \to \infty} \varepsilon^{(t)}$.

We consider two cases:

**Case 1: Finite-time convergence.** If $\varepsilon^{(t)} \leq \epsilon$ for some finite $t$, then the stopping rule triggers and we have safe convergence at $\tau^*$.

**Case 2: Asymptotic convergence.** Suppose $\varepsilon^{(t)} > \epsilon$ for all finite $t$. We show that $\varepsilon^* \leq \epsilon$.

By the bounded reliability assumption, there exists $\rho > 0$ such that $\max(r_H^{(t)}, r_L^{(t)}) \geq \rho$ for all $t$. This implies that the reliability penalty terms are bounded:

$$\beta_1(1 - r_L^{(t)}) + \beta_2(1 - r_H^{(t)}) \leq \beta_1 + \beta_2 - \min(\beta_1, \beta_2)\rho$$

Now, if $\varepsilon^* > \epsilon$, then the KL divergence terms cannot vanish asymptotically. But persistent disagreement would generate gradients that continue to decrease the energy, contradicting convergence. Therefore, we must have $\varepsilon^* \leq \epsilon$.

The reliable agent dominates the consensus because its predictions carry higher weight in the energy minimization process. $\square$

*Proof of Theorem 2.* If $r_H^{(t)}, r_L^{(t)} \to 0$, then the reliability penalties satisfy:

$$\liminf_{t \to \infty} [\beta_1(1 - r_L^{(t)}) + \beta_2(1 - r_H^{(t)})] = \beta_1 + \beta_2$$

Thus, for any $\epsilon_{\text{low}} < \beta_1 + \beta_2$, there exists $T$ such that for all $t \geq T$:

$$\varepsilon^{(t)} > \epsilon_{\text{low}}$$

Moreover, as reliabilities vanish, the gradients diminish because:

$$\|\nabla \varepsilon\| = O\left(\max(r_H^{(t)}, r_L^{(t)})\right) \to 0$$

This implies $\Delta \varepsilon^{(t)} \to 0$. The stopping condition in Eq. 1 detects this stagnation above $\epsilon_{\text{low}}$ and triggers termination. $\square$

*Proof of Corollary 1.* For $N$ agents, the energy functional is:

$$\varepsilon^{(t)} = \sum_{i \neq j} \alpha_{ij} D_{KL}(P_i^{(t)} \| P_j^{(t)}) + \sum_{i=1}^{N} \beta_i(1 - r_i^{(t)})$$

The convexity of KL divergence and linearity of reliability terms ensure that the multi-agent energy inherits the smoothness properties of the two-agent case. The proofs extend by considering the worst-case pairwise disagreement and the maximum reliability among agents.

Specifically, if there exists a reliable subgroup (agents with $r_i^{(t)} \geq \rho > 0$), they will dominate the consensus. If all reliabilities vanish, the energy remains bounded away from zero and descent stagnates, triggering the stopping condition. $\square$

## C EXPERIMENTAL DATASETS

### C.1 SYNTHETIC DATASET

To simulate agents with varying capabilities and enable experimental simulations across multiple scenarios, we assess their performance via a series of simulations. Below, we detail the architecture of our simulation framework:

**Synthetic Data Generator** The sample generation process is based on a predefined set of ground truth rules, as shown in Table 2. Initially, the generator selects a set of labels based on the rule weights, simulating population-level decision-making processes. This selection is formalized by the equation:

$$k \sim \text{Mult}\left(\text{softmax}\left(\text{sigmoid}^{-1}\left(\boldsymbol{w}_1^\top \boldsymbol{\phi}_1(\boldsymbol{x}), \ldots, \boldsymbol{w}_k^\top \boldsymbol{\phi}_k(\boldsymbol{x})\right)\right)\right),$$

where $\boldsymbol{w}_1^\top \boldsymbol{\phi}_1(\boldsymbol{x})$ represents the feature functions associated with the rule set, and the labels are selected if the corresponding features satisfy the rule conditions. For a valid sample, at least one of

the rules corresponding to the selected labels must be satisfied, while none of the rules associated with other labels should hold. In our simulation, label $k_0$ represents a rare class governed by longer rules, while labels $k_1$ and $k_2$ correspond to common classes with simpler criteria.

We divided the entire dataset into two disjoint subsets: (i) a training dataset $\mathcal{D}_t$ with 20000 samples and (ii) an evaluation dataset $\mathcal{D}_e$ with 100 samples. The training dataset $\mathcal{D}_t = \{\boldsymbol{x}_t, \{y_l\}_{l=1}^L, y_t, r_t, y^*\}$ includes feature vectors, class labels, and rule-level annotations. The evaluation dataset $\mathcal{D}_e$ is exclusively reserved for evaluation purposes.

Table 2: The ground truth rule set.

| Label | Rules | Weight |
|-------|-------|--------|
| $k_0$ | 1: $x_0 \wedge x_1 \wedge \neg x_2 \wedge x_3$ | 1.5 |
|       | 2: $x_3 \wedge x_4 \wedge x_7 \wedge \neg x_9$ | 1.5 |
| $k_1$ | 3: $x_3 \wedge x_4 \wedge x_5$ | 1.4 |
|       | 4: $x_6 \wedge x_7 \wedge x_9$ | 1.6 |
| $k_2$ | 5: $x_1 \wedge x_3 \wedge x_4$ | 1.7 |
|       | 6: $x_4 \wedge x_7 \wedge x_9$ | 1.3 |

**Agent Simulator**    In our framework, each agent is modeled as a rule-based probabilistic decision-maker. Each agent is equipped with a unique set of rules, characterized by specific conditions, classes, and weights. These rule sets may deviate from the ground-truth rules, reflecting heterogeneous expertise and potential biases. Unlike deterministic classifiers, the agents select actions according to the probability distribution given by the softmax function over the rule weights.

To evaluate collaborative decision-making under diverse conditions, we design several experimental scenarios, each consisting of a pair of agents with distinct rule sets and weight assignments. Table 3 shows representative configurations.

## C.2    Real-world Datasets

We also evaluate our framework on three publicly available classification datasets: **Drug Classification**, **Weather Type Classification**, and **Customer Segmentation**.

- **Drug Classification**: This dataset contains **200 samples** with **6 features**: Age, Sex, Blood Pressure (BP), Cholesterol level, Na-to-K ratio, and an identifier. The target variable is the prescribed **Drug type**, a categorical label with **5 classes** (Drug A, Drug B, Drug C, Drug X, Drug Y).

- **Weather Type Classification**: This dataset provides meteorological information for weather condition recognition. It consists of approximately **13,000 samples**, each described by **11 features**, including Temperature, Humidity, Wind Speed, Precipitation, Atmospheric Pressure, UV Index, Visibility, and categorical attributes such as Cloud Cover, Season and Location. The target label is **Weather Type**, a categorical variable with **4 classes**: Rainy, Sunny, Cloudy, and Snowy.

- **Customer Segmentation**: This dataset contains over **8,000 customer records**, with **8 features** such as Gender, Age, Education Level, Profession, Work Experience, Spending Score, Family Size, and Var_1. The target label is **Segmentation Class**, denoted as A, B, C, D.

Table 3: The rules assigned to each rule-based decision-maker in different scenarios.

| Scenario | Model | Rule Set | Weight |
|---|---|---|---|
| Scenario 1 Ideal | Agent 1 | $a_1 \leftarrow x_3 \wedge x_4 \wedge x_5$ | 1.4 |
| | | $a_1 \leftarrow x_6 \wedge x_7 \wedge x_9$ | 1.6 |
| | | $a_2 \leftarrow x_1 \wedge x_3 \wedge x_4$ | 1.7 |
| | | $a_2 \leftarrow x_4 \wedge x_7 \wedge x_9$ | 1.3 |
| | | $a_0 \leftarrow x_3 \wedge x_4$ | 1.3 |
| | Agent 2 | $a_0 \leftarrow x_3 \wedge x_4 \wedge x_7 \wedge \neg x_9$ | 1.5 |
| | | $a_0 \leftarrow x_0 \wedge x_1 \wedge \neg x_2 \wedge x_3$ | 1.5 |
| | | $a_1 \leftarrow x_3 \wedge x_4$ | 1.5 |
| | | $a_2 \leftarrow x_1 \wedge x_3$ | 1.5 |
| Scenario 2 One Strong & One Weak | Agent 1 | $a_1 \leftarrow x_3 \wedge x_4 \wedge x_5$ | 1.4 |
| | | $a_1 \leftarrow x_6 \wedge x_7 \wedge x_9$ | 1.6 |
| | | $a_2 \leftarrow x_1 \wedge x_3 \wedge x_4$ | 1.7 |
| | | $a_0 \leftarrow x_3 \wedge x_4 \wedge x_7 \wedge \neg x_9$ | 1.5 |
| | | $a_0 \leftarrow x_0 \wedge x_1 \wedge \neg x_2 \wedge x_3$ | 1.5 |
| | | $a_2 \leftarrow x_1 \wedge x_3$ | 1.2 |
| | Agent 2 | $a_2 \leftarrow x_4 \wedge x_7 \wedge x_9$ | 1.7 |
| | | $a_0 \leftarrow x_3 \wedge x_4$ | 1.5 |
| | | $a_1 \leftarrow x_3 \wedge x_4$ | 1.5 |
| | | $a_2 \leftarrow x_1 \wedge x_3$ | 1.3 |
| Scenario 3 Noisy- complementary | Agent 1 | $a_1 \leftarrow x_3 \wedge x_4 \wedge x_5$ | 1.4 |
| | | $a_1 \leftarrow x_6 \wedge x_7 \wedge x_9$ | 1.6 |
| | | $a_2 \leftarrow x_1 \wedge x_3 \wedge x_4$ | 1.7 |
| | | $a_0 \leftarrow x_3 \wedge x_4$ | 1.3 |
| | | $a_2 \leftarrow x_1 \wedge x_3$ | 1.3 |
| | Agent 2 | $a_2 \leftarrow x_4 \wedge x_7 \wedge x_9$ | 1.3 |
| | | $a_0 \leftarrow x_3 \wedge x_4 \wedge x_7 \wedge \neg x_9$ | 1.5 |
| | | $a_0 \leftarrow x_0 \wedge x_1 \wedge \neg x_2 \wedge x_3$ | 1.5 |
| | | $a_1 \leftarrow x_3 \wedge x_4$ | 1.3 |
| | | $a_2 \leftarrow x_1 \wedge x_3$ | 1.0 |
| Scenario 3 Noisy- noncomplementary | Agent 1 | $a_1 \leftarrow x_6 \wedge x_7 \wedge x_9$ | 1.6 |
| | | $a_0 \leftarrow x_3 \wedge x_4$ | 1.5 |
| | | $a_1 \leftarrow x_3 \wedge x_4$ | 1.4 |
| | | $a_2 \leftarrow x_1 \wedge x_3$ | 1.5 |
| | Agent 2 | $a_2 \leftarrow x_4 \wedge x_7 \wedge x_9$ | 1.3 |
| | | $a_0 \leftarrow x_3 \wedge x_4$ | 1.5 |
| | | $a_1 \leftarrow x_3 \wedge x_4$ | 1.4 |
| | | $a_2 \leftarrow x_1 \wedge x_3$ | 1.1 |

# D  LLM-BASED BEHAVIOR SIMULATION

---

**Complete Prompt Template for the Rule-Weighted Multi-Agent Classification Environment**

**Task Description**

- Each rule has the form "IF conditions THEN class = $c$".

- For a sample, sum the weights of all triggered rules per class; predict the highest score (ties broken by priority).

- Multiple agents have different prior strengths/weaknesses; this only affects their initial and adjustment tendencies on weights.

- **Data Type**:

  - If using real data, the background information about the dataset must be provided (e.g., source, nature of features, and any domain-specific considerations).
  - If using synthetic data, assume it follows the general distribution and properties defined by the experiment setup.

**Definitions and Constraints**

- **Features**: discrete or discretized attributes.

- **Classes**: the label set.

- **Rules**: logical predicates (AND/OR/NOT) implying a target class.

- **Rule weights**: real values in $[0, 2]$, contributing to class scores.

- **Agents**: each has a role description and class strengths/weaknesses.

- Update *weights only*; do not add/remove/modify rules or conditions.

- Keep rule IDs unchanged; prioritize fixing weak-class errors; push misleading rules toward 0.

- Output must be strict JSON with no explanations when requested.

**Input Format**

- `features` = {FEATURES}

- `classes` = {CLASSES}

- `agents` = {AGENTS} (`role_desc`, strengths/weaknesses)

- `initial_rules` = {INITIAL_RULES_BY_AGENT}

- `feedback_batch` = {FEEDBACK_BATCH} (optional)

- `command` = {"INIT" or "CONTINUE"}

**Your Task**

- Analyze rule–feature logic; detect conflicts/redundancy.

- Adjust weights within $[0, 2]$ using feedback; avoid large jumps and overfitting.

- Preserve all rule texts and IDs; resolve ties via preset priority.

- If `command="CONTINUE"`, return only updated weights and rule text.

**Output (strict JSON)**

```
{"current_rules": "<concatenate the exact rule text>",
  "rule_weights": {"rule1": 1.35, "rule2": 0.90}}
```

**Minimal Rule Example**

```
rule1: IF feature_3=1 AND feature_4=1 AND feature_5=1
THEN class=1.
rule2: IF feature_6=1 AND feature_4=1 AND feature_9=1
THEN class=1.
rule3: IF feature_1=1 AND feature_3=1 THEN class=0.
```

---

# E  STEPWISE CONFUSION MATRIX VISUALIZATION OF EACH SCENARIO

We present the confusion matrix variations for two scenarios in Fig. 3, and here we show the confusion matrix visualization results for all remaining scenarios and datasets, as shown in Fig. 5.

# F  BROADER IMPACT AND LIMITATION

Our framework provides a lightweight, theory-grounded tool for enhancing the safety of collaborative decision-making between humans, AI systems, and domain experts. By preventing harmful consensus, it has the potential to improve reliability in high-stakes fields such as medical diagnosis and scientific discovery, fostering more trustworthy and effective human-AI partnerships.

Despite proposing a natural multi-agent extension of the CEM framework in Section 2.6, including a defined multi-agent consensus energy function and outlined stopping-time/steering logic, this study has a limitation: existing theoretical analyses and empirical validations all focus exclusively on the two-agent setup. No rigorous mathematical proofs or systematic experiments support the multi-agent extension, and future work will supplement multi-agent theoretical proofs and design targeted experiments to validate its generality.

# G  THE USE OF LARGE LANGUAGE MODELS (LLMS)

Large language models (LLMs) were used in this work exclusively for polishing the writing and correcting grammar errors. All substantive research ideas, methodological design, and scientific conclusions presented in this paper were independently developed and validated by the authors.

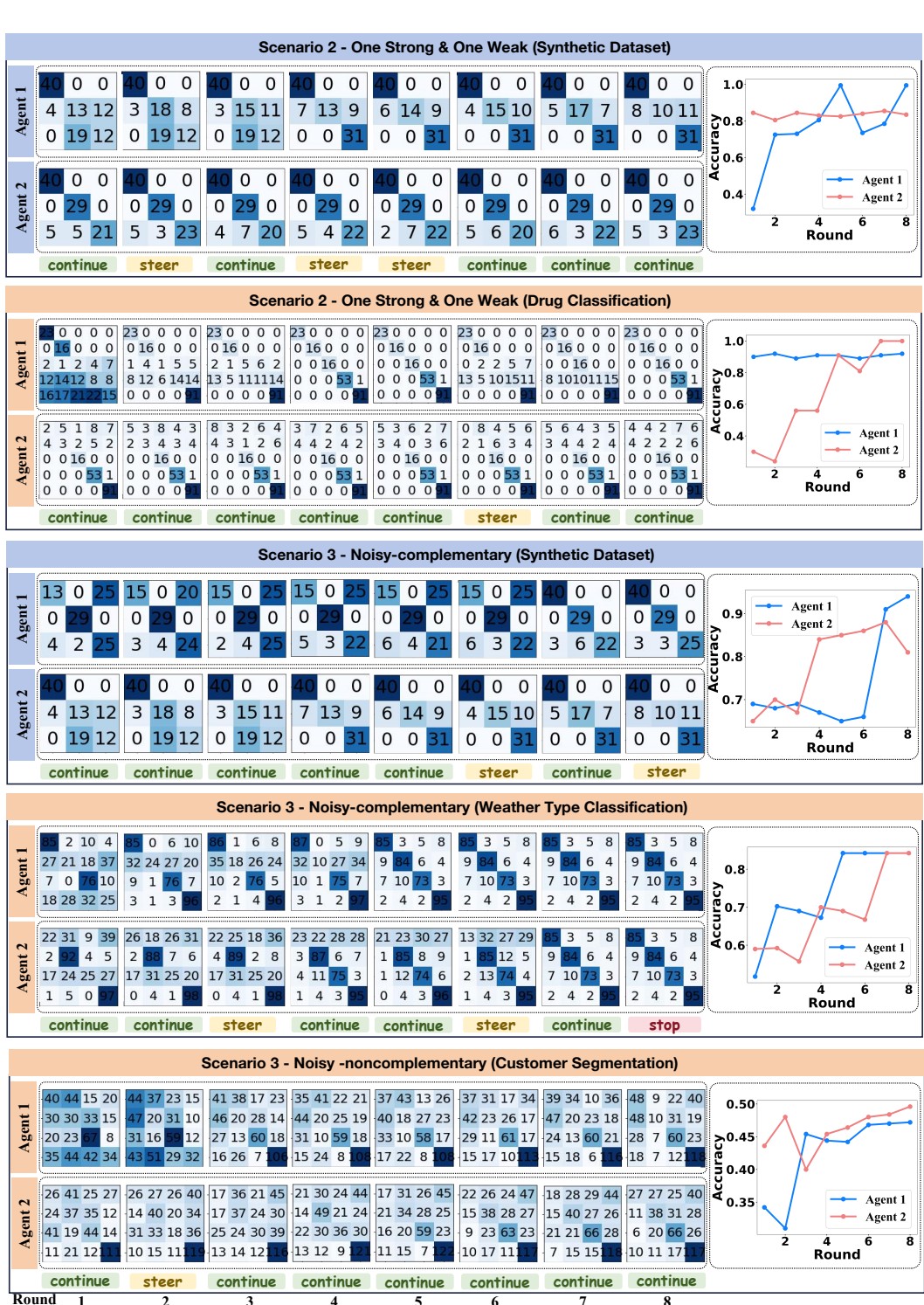

Figure 5: Stepwise confusion matrices and accuracy trends with monitor instructions.

