# OpenReview forum: "Consensus Energy Minimization: Ensuring Reliable Convergence in Collaborative Deliberation"
_ICLR.cc/2026/Conference — ICLR 2026 Conference Withdrawn Submission_

### Official Review · Reviewer_GXiu · 2025-10-30

**Soundness:** 2
**Presentation:** 2
**Contribution:** 2
**Rating:** 2
**Confidence:** 3

**Summary:**

This paper introduces Consensus Energy Minimization (CEM), a lightweight monitoring framework designed to regulate collaborative deliberation between heterogeneous agents (like humans and AIs). The goal is to prevent agents from prematurely agreeing on an unreliable conclusion.

**Strengths:**

while energy-based models for consensus are not new, the authors propose a novel confusion-aware consensus energy functional that innovatively combines a penalty for inter-agent disagreement (KL divergence) with a penalty for low-reliability consensus (derived from historical confusion matrices).

**Weaknesses:**

1. Agent heterogeneity limitation:

The paper's central motivation is to regulate deliberation among "heterogeneous agents," including humans and AI systems. However, the core methodology proposed appears to fundamentally contradict this claim in two major ways:

- Inapplicability to Human Agents: The paper explicitly names "Agent H (human or human-like)" as a target participant. Yet, the primary intervention mechanism, STEER , is defined as a "small projected gradient step" used to update a numerical feature-weight vector w_H​. The authors offer no mechanism or discussion for how this purely mathematical operation could be applied to a human's cognitive reasoning, rendering the human-AI collaboration claim unsupported by the method.

- Requires Homogeneous Feature Space: The STEER mechanism's objective function relies on minimizing the term $∣∣w_H​−w_{expert(t)}​∣∣^2$ . This is a squared Euclidean distance between the feature-weight vectors of the two agents. This calculation is only possible if both agents' reasoning is represented by vectors of the exact same dimension. This implies a high degree of homogeneity in the agents' architectures, which directly undermines the paper's goal of managing truly heterogeneous agents (e.g. a human and a deep network).

2. Practical and Theoretical Flaws in Confusion Matrix Usage:

The framework's reliability measure is based on an agent's confusion matrix $C_a$​. This component introduces two significant limitations:

- Practical Scalability: The paper suggests $C_a$​ can be "estimated from historical data or a calibration set". This is a practical approach for a small number of classes (K), but the $K\times K$ matrix becomes quadratically large and sparse as K increases, requiring an enormous calibration set to estimate accurately. This severely limits the method's applicability to complex, real-world classification tasks.

- Contradiction with Agent Updating: The methodology assumes a static confusion matrix based on "historical" performance. However, the STEER mechanism is explicitly designed to change the weaker agent's reasoning by updating its weights. An agent whose predictive model is actively being changed will, by definition, have a new and different confusion matrix. The framework provides no mechanism for updating Ca​ after a STEER operation, meaning all subsequent "Consensus Energy" calculations would be based on stale, incorrect reliability data. The fact that "Dynamic reliability" is listed as a future extension confirms this is a fundamental flaw in the current method.


3. Major Inconsistencies and Lack of Rigor in Presentation:

The paper suffers from significant inconsistencies and a lack of notational rigor.

- The paper's mathematical formalism is inconsistent. For example, Figure 1 is internally contradictory: It first presents the energy functional using $\beta_1$​ as a simple coefficient: $\epsilon = \dots + \beta_1(1-r_L)+\dots$. It then immediately re-defines $\beta_1$​ as a function of reliability on the next line: $\beta_1=a (1-r_L)$. Such inconsistencies suggests the paper was prepared in a rush and makes the core methodology ambiguous.

- Unclear Experimental Setup: The paper's experimental setup is not described with sufficient clarity for reproducibility. Furthermore, the paper provides no details on how these models were pre-trained or initialized, which is a critical omission. For instance, it's unclear if the models were trained on a dataset or simply initialized with pre-defined weights, as implied by the synthetic data setup.

**Questions:**

1. Your paper claims to work for "Agent H (human-like)". How do you apply the mathematical STEER operation, which is a "projected gradient step", to a human's cognition?

2. You are using an LLM to simulate the interactions. Is it generally possible to use an LLM as an agent in your setup? Given that the LLMs are essentially classifiers, but conditioned on a prefix of text.

3. Figure 4 is confusing. It plots "Accuracy" (left y-axis) and "Max Round" (right y-axis) in the same bar cluster, misleadingly grouping different units. Could you please clarify this presentation?

---

### Official Review · Reviewer_HztA · 2025-10-31

**Soundness:** 2
**Presentation:** 3
**Contribution:** 2
**Rating:** 4
**Confidence:** 3

**Summary:**

This paper proposes Consensus Energy Minimization (CEM) to regulate multi-agent deliberation. CEM minimizes an energy functional based on agent disagreement (KL divergence) and reliability (confusion matrices). A monitor uses this energy to CONTINUE, STOP, or STEER (adjusting weaker agents' feature weights via gradient descent).

**Strengths:**

1. Formalizing deliberation as energy minimization is principled, effectively capturing the trade-off between agreement and confidence.

2. Relevant Problem. CEM addresses the critical issue of unreliable consensus in collaborative AI with a lightweight approach.

3. Strong Efficiency in Simulation. Within the simulated environment, CEM demonstrates improved accuracy and efficiency (Fig 4).

**Weaknesses:**

1. This work fails to demonstrate real-world generalization. Datasets are small and limited to classification. Agents are partly LLM‑simulated and the confusion‑matrix estimation protocol is not fully specified.

2. There also seems no formal definition of safe consensus in terms of calibration between energy threshold and decision risk.

3. There is a large body of consensus optimization and mirror descent with KL/Bregman potentials work that may offer stronger guarantees (e.g., monotone operators/contractive mappings on the simplex).

**Questions:**

1. Please provide an ablation disabling STEER but retaining STOP/CONTINUE monitoring.

2. Justify why "natural deliberation" guarantees energy decrease?

3. What is the aggregated decision reported as the “final” output—majority vote, highest‑reliability agent, or something else?

---

### Official Review · Reviewer_bue2 · 2025-11-04

**Soundness:** 3
**Presentation:** 3
**Contribution:** 2
**Rating:** 4
**Confidence:** 3

**Summary:**

This paper proposed a new monitoring framework which regulates collaborative decision-making without requiring domain-specific supervisions. The proposed Consensus Energy Minimization (CEM) framework formalizes deliberation as a dynamical system. In such system, a confusion-aware consensus energy functional tracks both disagreement and convergence in low-reliability regions. To aid convergence to high-confidence consensus,  stopping-time rules are applied to halt or continue discussion toward an agent’s local expertise. Theoretically, the paper showed that CEM provably avoids harmful convergence and achieves stability in safe consensus regions. Empirical results further support such findings.

**Strengths:**

- Overall I found this work to be interesting and provided a new framework for reliable consensus convergence based on the energy minimization functional;
- the paper is well-organized and presented;
- the consensus energy function defined based on the KL divergence is intuitive and balances between the persistent disagreement among agents and the convergence to low-confidence outcome;
- empirical results provided a comprehensive set of experiments on synthetic and real-world data

**Weaknesses:**

while I found the framework to be interesting, I have some concern on the technical contributions:
- the proposed energy function involves multiple parameters to be tuned in practice, e.g. \alpha_1, \alpha_2 etc. The values of these coefficients can affect the outcome significantly. There is a lack of discussions on how they can be selected in practice.
- the energy function is based on the KL divergence and the confusion matrix. however the consution matrix may need to be estimated from samples instead of being known a priori. It is unclear if the energy minimization algorithm can be computed efficiently with unknown confusion matrices.
- the complexity of the algorithm grows exponentially as the number of agents increase as well, which makes it expensive to execute when the number of agents are large.

**Questions:**

-  Can you provide some discussion on the convergence rate in Theorem 1, e.g. under certain simpler scenarios?

---

### Official Review · Reviewer_6mb1 · 2025-11-05

**Soundness:** 2
**Presentation:** 2
**Contribution:** 2
**Rating:** 4
**Confidence:** 2

**Summary:**

This paper introduces consensus energy minimization, a monitoring technique to steer collaborative agents out of disagreement or agreement to a suboptimal convergent point. They define a consensus energy functional that captures the disagreement (KL-terms) and the reliability of the agent (which is derived using a running confusion matrix). The confusion matrix captures how many times the given agent predicted the right label for a ground-truth label. Using the monitoring framework they allow an iterative process of interaction between agents, where after each prediction they compute reliability, decide whether to stop, steer, or continue consensus, until convergence has been achieved (energy falls below a threshold). They provide theoretical justification and experimental evidence that their monitoring framework can improve reward perfo

**Strengths:**

1. Novel Framing - The treatment of deliberation among agents as a dynamical system with a monitored energy functional is elegant and conceptually interesting. I like the combination of divergence and reliability penalties in the energy definition.
2. Multi-Agent Extensibility - The authors explicitly mention how the framework can be extended to > 2 agent settings, as well as how to handle online settings.
3. Practical relevance - In domains where human + computer agents deliberate (the HCI paradigm), the risk of premature or misguided consensus is real. A lightweight monitor that doesn’t require retraining or heavy supervision is a compelling approach.

**Weaknesses:**

1. Limited Extensibility due to Reliability Measure - This approach relies upon 1) a measure of reliability that we can update over time 2) a formulation of a belief that decomposes into weights and the predictions per class. Therefore, while the experiments cover multiple domains, I feel the formulation is limited to classification tasks. How many agentic tasks are classification tasks? Is this method extensible for non-classification tasks and if so how?
2. Theoretical Brittleness - The theory relies upon a few assumptions, like the feature weight vectors belonging to a convex set (is this true based on the gradient updates to them during the method?). I would like the authors to justify these assumptions in the context of realistic problems that their method may be applied to.
3. Implementation Detail - There aren't any details of the exact large language models used in the experiment.
4. Figures - Typo on Figure 4. You used "Synthectic instead of synthetic".

**Questions:**

1. How do you imagine this method may be applied in a more general setting, where there isn't a way to verify the agents' success directly and update their feature weights?

---

### Note · Authors · 2025-12-04

I have read and agree with the venue's withdrawal policy on behalf of myself and my co-authors.